ecology

Goldilocks Principle, *Lophelia pertusa*, flow velocity, smoothed-particle hydrodynamics modelling, particle image velocimetry, coral

**Author for correspondence:**
S. J. Hennige
e-mail: s.hennige@ed.ac.uk

# Using the Goldilocks Principle to model coral ecosystem engineering

S. J. Hennige[1], A. I. Larsson[2], C. Orejas[3], A. Gori[4], L. H. De Clippele[1], Y. C. Lee[5], G. Jimeno[6], K. Georgoulas[1], N. A. Kamenos[7] and J. M. Roberts[1]

[1]Changing Oceans Group, School of GeoSciences, University of Edinburgh, Edinburgh, UK
[2]Department of Marine Sciences, Tjärnö Marine Laboratory, University of Gothenburg, Gothenburg, Sweden
[3]Instituto Español de Oceanografía, Centro Oceanográfico de Gijón, IEO, CSIC, Gijón, Spain
[4]Departament de Biologia Evolutiva, Ecologia i Ciències Ambientals, Universitat de Barcelona, Barcelona, Spain
[5]School of Engineering, Computing and Mathematics, University of Plymouth, Devon, UK
[6]School of Engineering and Physical Sciences, Heriot-Watt University, Edinburgh, UK
[7]School of Geographical and Earth Sciences, University of Glasgow, Glasgow, UK

SJH, 0000-0002-3059-604X; NAK, 0000-0003-3434-0807; JMR, 0000-0003-1688-5133

The occurrence and proliferation of reef-forming corals is of vast importance in terms of the biodiversity they support and the ecosystem services they provide. The complex three-dimensional structures engineered by corals are comprised of both live and dead coral, and the function, growth and stability of these systems will depend on the ratio of both. To model how the ratio of live : dead coral may change, the 'Goldilocks Principle' can be used, where organisms will only flourish if conditions are 'just right'. With data from particle imaging velocimetry and numerical smooth particle hydrodynamic modelling with two simple rules, we demonstrate how this principle can be applied to a model reef system, and how corals are effectively optimizing their own local flow requirements through habitat engineering. Building on advances here, these approaches can be used in conjunction with numerical modelling to investigate the growth and mortality of biodiversity supporting framework in present-day and future coral reef structures.

## 1. Introduction

Coral reef ecosystems represent one of the most structurally complex habitats in the oceans, and support biodiverse ecological communities spanning both tropical and deep-sea environments [1–7]. Key to this biodiversity support are the complex three-dimensional branching framework structures that many species of 'stony corals' (Scleractinia) create. Importantly, once a coral dies, this framework persists until it is bio-eroded, settled by other organisms or infilled with sediment, and the ratio of live : dead coral is an important determinant of biodiversity found on reefs. Due to the diverse communities they support, and the threats they face such as bleaching [8] and ocean acidification [9,10], considerable attention is now given to understanding the future state of coral reef assemblages. A key issue to understand is how stony corals grow in different conditions in the present day, as that will have a major bearing on the diversity of these ecosystems in the future. While coral growth has been successfully quantified for many coral species through experimentation, and communities can be characterized through *in situ* surveys, there exists a gap in being able to model how coral assemblages may grow, die and degrade at a habitat scale.

Existing models have explored how tropical corals grow at a fine scale, and have resulted in outputs very similar to real-world data, specifically from experimentally grown pieces of coral [11,12]. These elegant models have relied upon a combination of an accretive growth model and hydrodynamics, have demonstrated how flow could impact various coral physiological efficiencies [11,13], and have shown how corals can realistically branch and grow according to

key drivers [12,14]. However, there exists a challenge on how to scale up from this modelling to something relevant at a larger habitat scale. To achieve this on a realistic timescale and to avoid lengthy model runs, such a model would have to be based on simple principles in addition to being dynamic, i.e. being able to react to changes in its own environment. To determine whether it is possible to model coral growth at the scale needed, we test whether the Goldilocks Principle could be a governing rule in such a model, where coral reef habitats grow according to conditions that are 'just right' (optimal) and die when they are not (suboptimal). This is examined in a model cold-water coral (CWC) deep-sea ecosystem, which are classified as vulnerable marine ecosystems (VMEs) [15] and are found throughout many of the world's oceans up to 3000 m deep [16].

To model coral growth, energetic inputs need to be identified. For tropical corals, this takes the form of heterotrophic feeding by the coral polyp, and photosynthetic production and translocation of carbohydrates from their photosynthetic endosymbionts. For CWC that live beyond the photic zone, that do not have photosynthetic symbionts and are opportunistic feeders [17,18], heterotrophic acquisition of prey capture by the coral, such as zoo- and phytoplankton, as well as particulate and dissolved organic matter is their main source of energy [19–22]. Good heterotrophic food supply and a current velocity that facilitates prey capture are therefore key for the existence of CWC reefs [23,24], and highlight that current velocity is an ideal parameter to examine the role of the Goldilocks Principle in coral reef habitat engineering.

*Lophelia pertusa*, one of the most widespread CWC species in the Atlantic Basin [5,16], is found at sites with both high and low current velocities [25–27]. From two well-characterized reefs, the Mingulay Reef Complex in Scotland, and Tisler Reef in Norway, current speeds have been observed to vary from approximately 2 to approximately 50 cm s$^{-1}$ and peak at approximately 100 cm s$^{-1}$ in the case of Tisler Reef [28–30]. Paradoxically, the flow speed for optimal food capture is much slower than would be expected given the range of velocities recorded on *L. pertusa* reefs. The determining speeds suitable for coral growth are therefore likely determined by local hydrodynamics around small-scale coral features that baffle the faster current velocities [25,31]. Experimental studies have determined that the current speed for optimal food capture is approximately 2–6 cm s$^{-1}$ [32–34]. If the flow speed is too low, e.g. below 1 cm s$^{-1}$, which has been found to be just strong enough to prevent prey navigation [33], zooplankton can potentially evade capture. If the current flow is too fast, coral tentacles are swept back and coral is unable to catch its prey [32,35]. Current velocities therefore appear to dictate the efficiency of food capture, and ultimately the growth of coral and the structures that they form. Although specifics with regard to energetic value, optimal velocity and food quality will vary by prey type, size and capture mechanism [21,32,35], a Goldilocks Principle schematic can be applied to this situation (figure 1). In such a construct, optimal prey capture and hence coral growth will occur at a specific range of current velocities. Flow velocities above and below the optimum range will result in low prey capture rates, and limit coral growth. A survival threshold can also be applied, where if inadequate prey is captured over a period of time, the coral would not have adequate energy and would die.

Coral mortality will occur if conditions are suboptimal with regard to current speed and prey capture (figure 1).

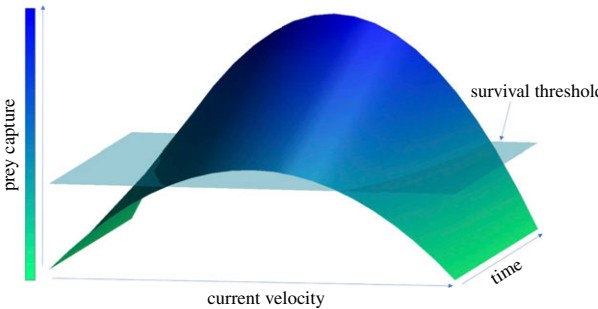

**Figure 1.** Curve of the theoretical prey capture rates of *L. pertusa* at different current velocities against time. Bisecting layer (rectangle) indicates a 'survival threshold' based on prey capture. Coral polyps not surpassing this threshold would die, leaving exposed 'dead' framework. Given adequate time, this modelled threshold can be surpassed either through prey capture in optimal or suboptimal conditions. (Online version in colour.)

This mortality is part of the normal reef growth process and can be observed on *L. pertusa* stony branching reefs worldwide, where live coral is growing on the top of the dead coral matrix with exposed skeletal framework [16,36] (figure 2). In these reefs, death of coral is likely driven by lack of available food or adequate current velocities [36]. This dead framework is crucial with regard to habitat provision on CWC reefs, as in addition to the dead coral skeleton framework supporting the live coral, it provides a settlement substrate for benthic organisms that cannot settle on live coral [3,38]. Understanding what controls the growth of live coral, and the proportion of live : dead areas of *L. pertusa* reefs would improve our prediction of the type of habitat provided by reefs under different flow regimes [36,39], and ultimately inform on the potential biodiversity supported by different habitats under present and projected future conditions. Other environmental variables that also control coral health or mortality can then be introduced, such as oxygen, temperature, salinity and pH [9,40,41], or other controls of growth like variability of light intensity if modelling growth in tropical or shallow coral species.

Here, we investigate how coral colonies modify their own hydrodynamic environment and subsequently grow by applying the Goldilocks Principle through two steps: (i) by quantifying the flow around coral reef colonies (*L. pertusa*) in a variety of scenarios using particle imaging velocimetry (PIV) to understand how coral framework can modify flow environment within a reef and (ii) using smooth particle hydrodynamics (SPH) [42,43] to provide a theoretical simulation of how this coral will grow when given simple rules that follow the Goldilocks Principle. Through combining PIV data with the introduction of SPH models and simple 'death rules', we also examine how energetic reserves would shape coral framework, and explore how the Goldilocks Principle could be used to investigate the live : dead ratios of the framework in both present-day and future scenario coral reefs.

## 2. Methods

### (a) Coral collection and flume tank preparation

Coral colonies for experimental PIV analysis in flumes tanks were collected in 107 m, Central Tisler reef (58°59.730′ N, 10°57.913′ E), using the remotely operated vehicle (ROV) Sperre SubFighter

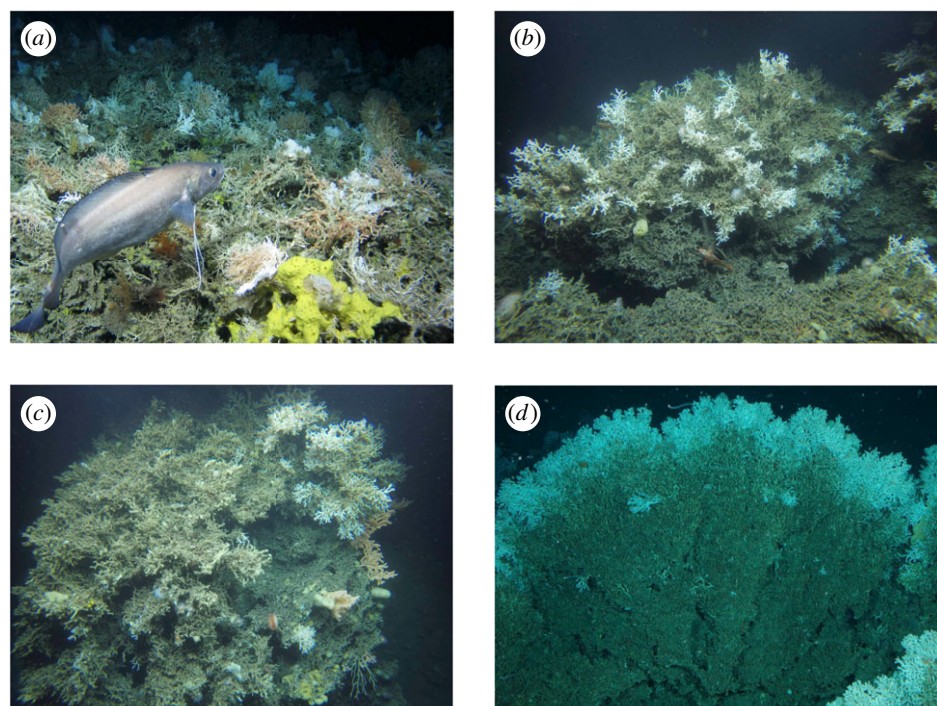

**Figure 2.** Examples of the complex three-dimensional matrix structure of cold-water coral reef frameworks in the NE Atlantic showing live coral (white/orange colour) and dead coral framework (grey). Live coral typically sits on the top of dead coral framework. (*a*) SE Rockall Bank (Changing Oceans 2012) and (*b*–*d*) Norwegian cold-water habitats imaged by JAGO in cruises POS525 and PS_ARK22-1a [37]. (Online version in colour.)

7500 DC deployed from the R/V Lophelia from Tjärnö Marine Laboratory (University of Gothenburg). *Lophelia pertusa* colonies ($n = 3$, $11.5 \pm 0.9$ cm height, $12.8 \pm 1.6$ cm diameter) and dead nubbins, which are defined here as small branches of *L. pertusa* with approximately five polyps ($n = 3$, $3.6 \pm 0.9$ cm height, $5 \pm 1$ polyps), were placed in a 7 m long and 0.5 m wide recirculating laboratory flume tank. The flume tank design produces a fully developed turbulent boundary layer at the working section, 5 m from the flume tank entrance [44]. Coral colonies and nubbins were placed in three different set-ups: (i) a coral nubbin fully exposed to input velocity, (ii) a coral nubbin protected by one coral colony located 5 cm in front of it, and (iii) a coral nubbin protected by two coral colonies, one 5 cm in front and another 8 cm behind (figure 3). The flume tank was filled to a water depth of 20 cm with seawater at 34.8 salinity maintained at 7.5°C.

## (b) Particle image velocimetry

Flow patterns in the areas behind *L. pertusa* colonies and around the coral nubbins were studied at average free-stream velocities of $2 \pm 0.15$, $5 \pm 0.3$, $10 \pm 0.3$ and $30 \pm 0.7$ cm s$^{-1}$ using PIV to match a variety of field-ranges. Recordings for each colony set-up were conducted both with and without the small nubbin present. An acoustic doppler velocimeter (ADV), by Nortek AS, measured the free-stream velocities at 10 cm above the flume tank floor. The water was seeded with 10 μm hollow glass spheres (Dantec Dynamics) and recordings were made using a system from LaVision with a 1600 × 1200 pixel camera (Imager Pro X) and a double-pulsed Nd:YAG laser (Litron, 30 mJ at 532 nm). The laser produced a 1 mm thick vertical light sheet parallel to the flow. The camera with a 24 mm Nikkor lens recorded images through the transparent sidewall of the flume tank covering an area of 13 × 10 cm at the centre of the working section where the corals were placed. For each combination of flow velocity and coral set-up, recordings were made in the double-frame mode at 10 Hz for 2 min (1200 recordings). Images were analysed with the DaVis 8.2.0 software (LaVision) using cross-correlation and multipass options with a final size

of 32 × 32 pixels at 50% overlap between interrogation windows. This set-up resulted in a resolution of 8 flow vectors cm$^{-1}$. To examine the ambient flow environment at nubbin tips, we used results from the recordings without nubbins present (only larger coral colonies were present) and extracted the instantaneous flow vectors at the positions of the nubbin tips. This was carried out for each combination of colony set-up and nubbin, resulting in three time-series of flow vectors extracted for each colony set-up (spatially at the positions of the nubbin tips). The vector series were further processed by calculations of instantaneous magnitudes of flow velocities and the average thereof. To compare flow velocities at coral nubbin tips for each treatment (exposed nubbins and set-ups with nubbins behind/in between colonies), separate one-way ANOVA with the Bonferroni *post hoc* tests were used for each of the four free stream velocities tested. These values were used to inform resultant SPH models.

## (c) Smoothed-particle hydrodynamics

A fully Lagrangian numerical SPH approach was employed to model and capture the dynamic and complex hydrodynamic fluid–solid interactions of coral growth. Conceptually, the SPH method operates using integral interpolation theory to discretize a set of partial differential equations into meshless points that are subsequently solved iteratively using an appropriate time integration scheme [45,46]. The meshless characteristics of the methodology innately conserves mass and allows natural tracking of solid–liquid interfaces which is readily extended to dynamically respond to newly imposed boundary conditions for coral growth.

The case of a newly settled single hemispherical coral is considered at the centre of the seabed (see electronic supplementary material, figure S1), where the coral's growth is predicted using a set of simple rules; the coral colony will grow towards surrounding regions where the hydrodynamic flow conditions are favourable within the optimum reported range [32–34]. As the coral grows it alters the hydrodynamics around it, modifying the dynamic flow environment for itself, and as a consequence

of that growth, to its surroundings. Optimum current flow conditions around the corals are the metric used to determine coral growth, and it is assumed that when that condition is met that corals are able to capture and ingest prey, providing them with energy for growth. Likewise, when flow conditions are undesirable, a death rule is implemented to ascertain when the above rules no longer apply, and a living coral particle will 'die'. This 'death rule' can be modified to allow corals to survive for a variable amount of simulated growth time in suboptimal flow conditions, and by doing so, simulate the use of its energetic reserves by the corals. No arbitrary branching rules were applied in the model, and any branching observed is a result of the dynamic interaction of the flow and the set of prescribed rules as outlined above.

An in-house developed SPH solver that has previously been developed to simulate single-phase flows and had its output compared to a finite volume solver [47] is used here to solve the following conservation equations for mass and momentum, expressed in the Lagrangian form:

$$\frac{D\rho}{Dt} = -\rho \nabla \cdot v \tag{2.1}$$

and

$$\frac{Dv}{Dt} = \frac{-1}{\rho} \nabla p + \frac{1}{\rho} (\nabla \cdot \tau) + \frac{F}{\rho}, \tag{2.2}$$

where $D\rho/Dt$ is the total or material derivative, $\nabla$ is the gradient, $p$ is the pressure, $v$ is the velocity, $t$ is the time, $\rho$ is the density, $\tau$ is the viscous stress tensor and $F$ is the body force per unit volume that typically includes contributions such as gravity and boundary forces. The weakly compressible [45] form of the SPH formulation employed here allows the pressure to be uniquely determined from the density field via an equation of state (EOS), and reduces the computational resources required to compute the pressure field without the need to resolve the pressure Poisson's equation for incompressibility [48]. Equations (2.1) and (2.2) are closed using Tait's EOS, given by

$$p = B \left[ \left( \frac{\rho}{\rho_0} \right)^{\gamma} - 1 \right], \tag{2.3}$$

where $\gamma = 7$ for water, $\rho_0$ is the reference density and $B$ is the reference pressure constant

$$B = \frac{\rho_0 c_s^2}{\gamma}, \tag{2.4}$$

with $c_s$ denoting the reference speed of sound chosen, usually between 10 and 100 times of the system's maximum velocity, so that density variations are limited to 1% of the reference density [45]. A value of $c_s = 10 \text{ m s}^{-1}$ was chosen for the simulations performed in this study. Numerical SPH discretization of equations are derived using a smoothed approximation of continuous formulations via an integral interpolation of surrounding Lagranian points or particles with a kernel function, $W$, that must satisfy the following conditions for:
symmetry

$$W(r, h) = W(-r, h), \tag{2.5}$$

limit

$$\lim_{x \to 0} W(r, h) = \delta(r), \tag{2.6}$$

and unity

$$\int_{\Omega} W(r, h) \, dr = 1, \tag{2.7}$$

where $r$ is the position vector, $h$ is the smoothing length of the kernel and $\delta$ is the Dirac $\delta$ function in the continuous domain,

$\Omega$. The Wendland kernel [49,50] is employed in the present study due to its superior dissipation characteristics at both the low and high Reynold number flows that prevent clustering effects from noisy vorticity fields

$$W(q, h) = \alpha \begin{cases} \left(1 - \frac{q}{2}\right)^4 (1 + 2q) & 0 \le q \le 2 \\ 0 & 2 > q, \end{cases} \tag{2.8}$$

with $q = |r|/h$ is the dimensionless kernel smoothing length ratio and $\alpha = 7/4\pi$ for two-dimensions problems.

In SPH formalism, the continuity equation (2.1) for fluid flow is rewritten in the following form:

$$\frac{D\rho}{Dt} = \sum_j^N m_j v_{ij} \cdot \nabla_i W_{ij}. \tag{2.9}$$

No additional artificial viscosity model is adapted. This work includes relatively small velocities and the particles fill all the numerical domain; therefore, a more realistic form of viscosity is adapted, as suggested by Morris et al. [51]. According to this, the momentum equation (2.2) for fluid flow is rewritten and solved in the following form:

$$\frac{Dv_i}{Dt} = -\sum_j^N m_j \left( \frac{p_j}{\rho_j^2} + \frac{p_i}{\rho_i^2} \right) \cdot \nabla_i W_{ij}$$
$$+ \sum_j^N m_j \left( \frac{\mu_i + \mu_j}{\rho_i \rho_j} \right) v_{ij} \left( \frac{1}{r_{ij}} \frac{\partial W_{ij}}{\partial r_{ij}} \right) + \frac{F}{\rho_i} \tag{2.10}$$

Here, the subscript $i$ represents the fluid particle of interest, $j$ is the neighbouring $N$ fluid particles and $ij$ is the difference in the value between particles $i$ and $j$. The mass and dynamic viscosity of the fluid is denoted by $m$, and $\mu$, respectively. The distance between particles $i$ and $j$ is denoted by $r_{ij}$.

After solving the continuity equation (2.9), a density correction algorithm is applied according to Ozbulut et al. [52]. In weakly compressible SPH, the pressure of particles is calculated using an artificial EOS and it is directly connected to the particle's density. Therefore, a density smoothing algorithm helps to avoid large density variations in the domain that can lead to numerical instabilities and inaccuracies using:

$$\tilde{\rho}_i = \rho_i - \varepsilon \sum_{j=1}^N \frac{m_j (\rho_i - \rho_j) W_{ij}}{0.5(\rho_i + \rho_j)} \tag{2.11}$$

where $\tilde{\rho}_i$ is the corrected density and $\varepsilon$ is an XSPH-like factor as proposed by Violeau [46]. For this work, the value of $\varepsilon$ is chosen to be equal to 0.01.

The system of equations for each fluid particle are integrated over time using the Verlet time integration scheme. The size of the time or growth-step is chosen by considering the Courant–Fredrichs–Lewy (CFL) condition to ensure that the maximum rate of numerical interaction propagation does not exceed the physical rate [53], and is coupled with two additional growth-step restrictions to account for viscous dissipation and body forces [45].

The seabed and coral particles are treated as solid surfaces using dynamic boundary conditions [54]. These boundary particles are solved using the same equations as the moving fluid particles, but their positions and velocities remain unaltered in time or are externally prescribed [46,47]. A fluid domain $10 \times 5 \text{ m}^2$ in size is employed to predict the growth of a coral seed (electronic supplementary material, figure S1) with steady-state parabolic inlet flow conditions on the left boundary, and an outlet on the right. A flow velocity of $50 \text{ cm s}^{-1}$, to simulate a typically fast-moving current that CWC are subjected to, is defined at the upper boundary and is located at a height to ensure that it does not influence the flow around the developing coral. A total of 20 000 SPH fluid particles were used and this corresponds to initial particle size, $\Delta x = 0.05 \text{ m}$. It should be noted

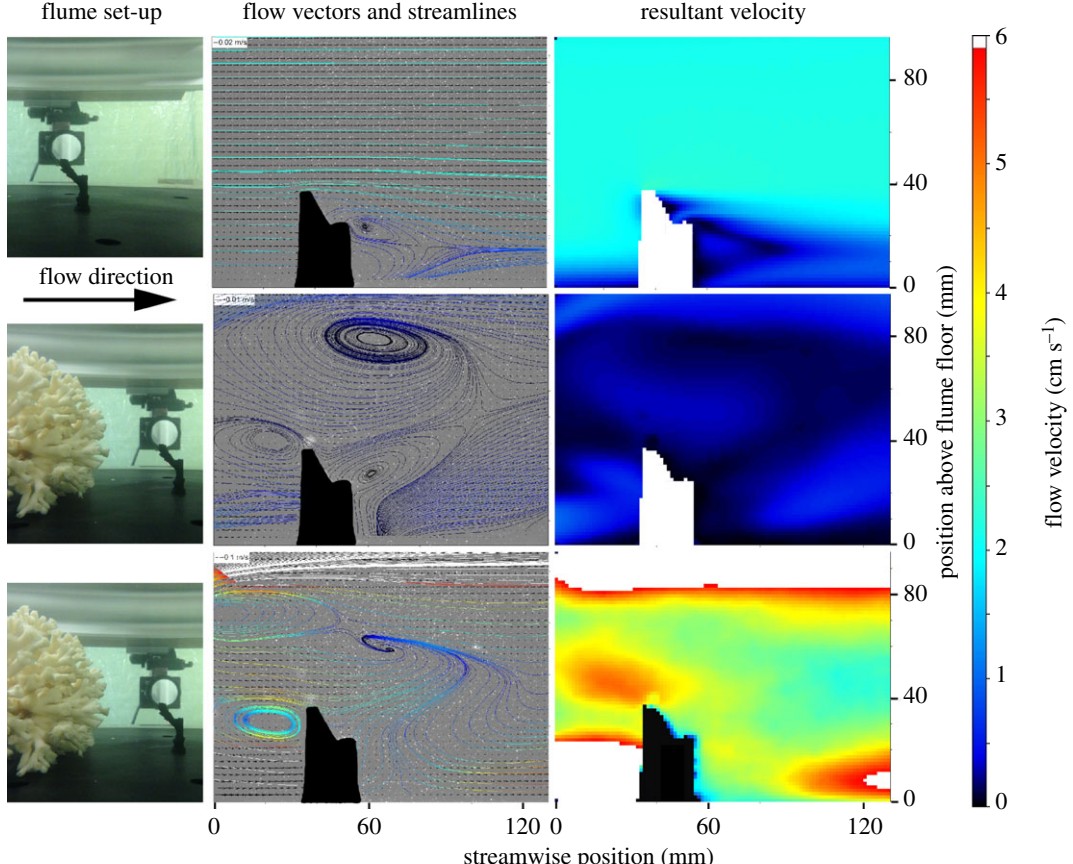

**Figure 3.** Experimental set-up and visualized disruption of water flow by corals. Left panels demonstrate experimental set-ups in flume tanks with *L. pertusa* colonies and nubbins. Middle and right panels demonstrate average flow velocity vector fields and velocity magnitudes around *L. pertusa* nubbins during 2 min PIV recordings at 10 Hz. Top row—nubbin exposed to 2 cm s$^{-1}$ free-stream flow. Middle row—nubbin placed behind a larger colony exposed to 2 cm s$^{-1}$ free-stream flow. Bottom row—nubbin located behind a larger colony exposed to 30 cm s$^{-1}$ free-stream flow. The solid black/white areas indicate no available data. (Online version in colour.)

that the simulated time in the model does not represent real physical time, and is only used as a means to advance the solution over successive iterative time- or growth-steps. The goal here is to predict the growth of coral based on a set of given simple rules that corals grow in conditions that are 'just right' (the Goldilocks Principle).

The underlying mechanism for coral growth was controlled by analysing the average local steady-state flow velocities of fluid particles adjacent to the coral particles such that it extends in the direction where velocities fall within the optimal range of 3–6 cm s$^{-1}$ and a proximity of $1.5 \cdot \Delta x$. Where the aforementioned condition for coral growth is met, the fluid particle is converted into a live coral particle to simulate coral growth. There are no additional particles inserted or deleted from the numerical domain. By converting the qualifying fluid particles into coral particles, the total number of particles in the domain remains constant at all times to avoid numerical instabilities (see electronic supplementary materials, Pseudocode supplementary code 1). This growth function is initiated at fixed growth-steps once the local flow is steady. A similar mechanism is applied using the death rule, where live coral particles are converted to dead coral particles when flow conditions are suboptimal over a pre-defined interval.

## 3. Results

### (a) Flow around corals: particle image velocimetry
The presence of coral nubbins created rapid decreases in flow velocities immediately in its wake (figures 3 and 4), leading to downstream areas of suboptimal flow of approximately 1 cm s$^{-1}$. The different treatments with and without coral colonies in flume tanks significantly affected the flow conditions at the nubbin tip at maximum height for all free-stream velocities (one-way ANOVAs $F_{4,10} = 343$, $p < 0.001$) (figure 3 and table 1). Compared to fully exposed coral nubbins, the presence of larger colonies at the front of the coral nubbin decreased flow velocities to the nubbin tip location (Bonferroni *post hoc*s, $p < 0.001$). However, there was no significant difference between resultant velocities at nubbin locations when between colonies (Bonferroni *post hoc*s, $p > 0.14$) (figure 4 and table 1). For nubbins located behind or in between other colonies, an input free-stream velocity of 30 cm s$^{-1}$ was needed for the nubbin to receive velocities adequate for prey capture (table 1).

### (b) Modelling coral growth: smoothed-particle hydrodynamics
The developed SPH model successfully simulated coral growth when current flow conditions in its vicinity are optimum and created branching coral structures that grew into a larger coral framework (figure 5; electronic supplementary material, figures S2 and S3). When no death rule was applied, the coral continued to grow into a rugose hemispherical shape (figure 5). When a death rule was applied, the coral particle died if flow velocities at its vicinity were suboptimal which, over time, resulted in significant dead regions of the

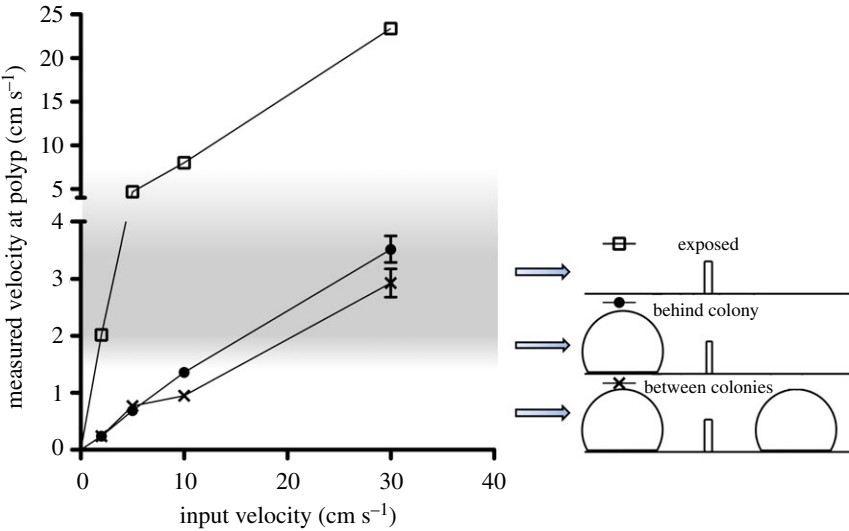

**Figure 4.** Water flow velocities at *L. pertusa* nubbin tips in flume experiments ± s.e. 'Exposed' nubbins were in full boundary layer flow, 'behind colony' nubbins had one colony placed in front of them and 'between colonies' had one colony in front and one behind it. The grey area indicates the broad region of optimum flow speeds for *L. pertusa* capture of zooplankton (approx. 2–6 cm s$^{-1}$ [32–35]).

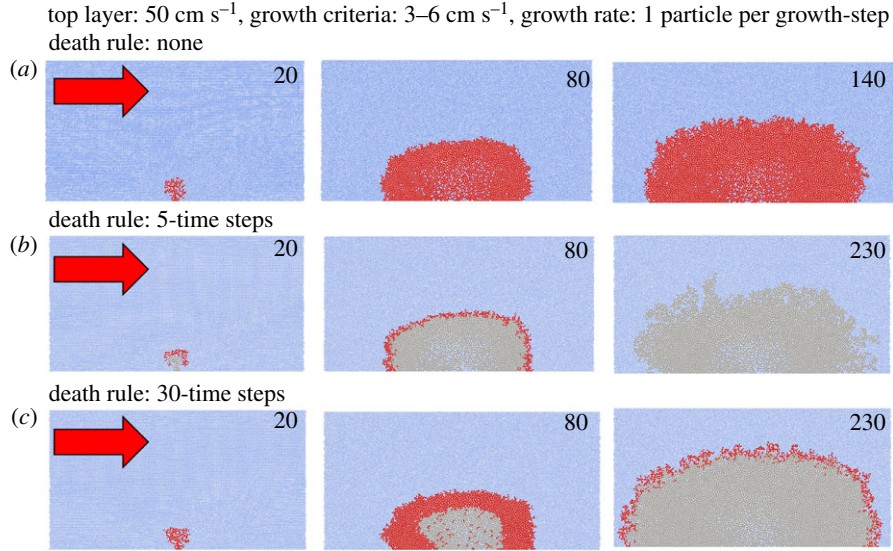

**Figure 5.** SPH models of coral growth. Top layer flows from the left to right side at 50 cm s$^{-1}$ as indicated by the red arrows. Growth rates of coral were one particle for every growth-step if steady-state surrounding velocities were optimal between 3 and 6 cm s$^{-1}$. Live coral is denoted in red, dead coral is grey and water particles in blue. Numbers in each of the three panels indicate simulation growth-step (e.g. 20, 80, 140, 230). (*a*) Coral growth with no death rule applied, (*b*) coral growth with a death rule applied if coral does not experience optimal flow for five growth-steps. (*c*) Coral growth with a death rule applied if coral does not experience optimal flow for 30 growth-steps. (Online version in colour.)

**Table 1.** Flume tank free-stream velocities and resultant flow velocities at nubbin tip locations in PIV experiments. Conditions were (i) fully exposed to input velocity, (ii) 5 cm behind a larger coral colony, and (iii) between two larger coral colonies, 5 cm behind the first and 8 cm in front the second. Italicized cells indicate a flow velocity within the broad optimal range for prey capture (approx. 2–6 cm s$^{-1}$ [32,33,35]).

| | flow velocity at nubbin tip (cm s$^{-1}$) | | | | | |
| --- | --- | --- | --- | --- | --- | --- |
| | **exposed** | | **behind colony** | | **between colonies** | |
| **input flow velocity (cm s$^{-1}$)** | **polyp area** | **range** | **polyp area** | **range** | **polyp area** | **range** |
| 2 | *2.02 (0.03)* | *1.97–2.07* | 0.24 (0.03) | 0.15–0.42 | 0.24 (0.01) | 0.23–0.26 |
| 5 | *4.70 (0.07)* | *4.59–4.82* | 0.69 (0.04) | 0.56–0.96 | 0.77 (0.04) | 0.70–0.84 |
| 10 | 8.02 (0.16) | 7.75–8.32 | 1.36 (0.07) | 1.18–1.68 | 0.95 (0.09) | 0.83–1.12 |
| 30 | 23.4 (0.50) | 22.6–24.3 | *3.52 (0.23)* | *2.93–4.34* | *2.93 (0.25)* | *2.51–3.38* |

coral reef framework. This was clearly shown when the death rule was applied at intervals of five simulation growth-steps, where a large region at the bottom and interior of the coral framework died leaving only live coral sitting on the top of dead coral at growth-step 80. As the model progressed, the entire coral colony eventually died due to changes in hydrodynamics leaving live coral exposed to periods of suboptimal velocities that meet the 'death rule' criteria (see growth-step 230, figure 5b). Once the death rule was modified to allow the coral to survive longer without meeting the growth criteria (30 growth-steps), coral colonies continued to grow, with a greater proportion of live coral surviving compared to the former simulation (figure 5c). In general, the results showed that increasing the death rule interval corresponded to more live coral at the end of the simulation (electronic supplementary material, figure S2).

An example of the velocity flow field calculated from the developed SPH solver is shown in electronic supplementary material, figure S3. The figure demonstrates that the methodology successfully captures typical flow profiles expected over coral structures, similar to those observed in figure 3. The velocity flow field delineates that the flow velocity adjacent to the coral is mostly small, with gradual changes in gradients that increase with its height. This is exemplified in electronic supplementary material, figure S3b, where the optimal flow velocities are identified in relation to the coral. The size and position of downstream eddies depend on the incident velocity and shape of the coral, allowing for evaluation of growth in areas 'hidden' from direct incident flow velocities.

## 4. Discussion

### (a) Flow around corals

Our data outputs clearly indicate that reef-forming coral can substantially modify their surrounding flow environment (figures 3 and 4; electronic supplementary material, figure S3 and table 1), and as structural complexity increases, so does the impact on local hydrodynamics. This will have significant implications for corals existing next to larger structures and can mean that flow speeds become suboptimal as the coral structure grows. This highlights the need of a tool such as SPH to simulate coral growth, as the optimal areas within the model will change in each iteration depending on new coral growth. However, it is important to note that most CWC reef systems do not experience unidirectional flow, and may be subjected to tidal regimes, downwelling currents (e.g. Mingulay Reef Complex), cascading with regard to submarine canyons, or residual flows lasting several days to weeks [27,28,30,55–58]. These variable flows would ensure that hydrodynamic conditions would vary over a reef area, and as current conditions change, some coral colonies will move from suboptimal to optimal conditions, and vice versa. This gives rise to the diverse and complex framework of coral colonies observed on CWC reefs (figure 2).

### (b) Modelling coral growth

The flow (SPH) models demonstrate the growth of a theoretical coral framework based on simple principles, and that the Goldilocks Principle can be applied to coral growth. When coral framework was formed with only one rule (grow where water flow conditions are optimal), coral was dense with high rugosity on the outer edges (figure 5a). The model was stopped after 140 simulated growth-steps, as growth was rapid and the coral outgrew the confining space of the model. No branching rule was applied to the model, so the framework grew simply where flow was optimal. The resultant model output raises interesting questions as to the control of L. pertusa branching, and what is environmentally, geographically or genetically controlled (i.e. how often do they bud and branch [59], recognized in terrestrial analogues [60,61]). The dense framework created in the model can also be observed on L. pertusa reefs and mounds, as older living branches thicken and fuse into each other [5,59,62]. However, our first model (figure 5a) was not representative of the quantity of dead coral framework evident off reefs. To account for this, we introduced a 'death rule' where coral would die if it experienced suboptimal flow conditions (and hence reduced prey capture) for a number of model growth-steps. If the death rule was too short (five growth-steps), the coral framework grew to a point where it created suboptimal velocities over its entirety, and total death of the colony occurred (figure 5b). When the death rule was increased (30 growth-steps), the resultant framework was mostly dead coral, with a veneer of live coral around the outside (figure 5c). This output is similar to what is often observed on L. pertusa reefs (figure 2), where a large proportion, at least 70% in some cases [36], of the visible framework is dead coral.

The introduction of the death rule, where coral framework survives as a whole as long as the death rule is not too short, supports that corals have energetic reserves that can be used to sustain metabolism if conditions are suboptimal. The ability of corals to use this store (and to what extent of a store they have) will depend on a variety of factors, such as prior food provision, environmental factors and reproductive state, given that reproduction is an energetically demanding process [63,64]. Previous studies on the energetic budgets of L. pertusa have indicated that their energetic reserves may sustain them under suboptimal conditions over a period of months [64,65]. The threshold for coral survival depends in part upon these energetic reserves, as that dictates the timescale corals can survive in suboptimal conditions. However, this will be further affected by the metabolic demand of the coral and any stressors that can impact this [63,66]. The variable amounts of dead framework at different reefs highlight the complexity of this question, as the extent of the colony will also be affected by additional factors [36]. Being able to model how corals grow and die in models such as those presented here represents an important first step in understanding as well as predicting the biodiversity provision that reefs under different flow regimes could provide.

### (c) Using the Goldilocks Principle as a governing principle for future coral growth modelling

The data presented support that cold-water coral framework grows according to the Goldilocks Principle, and highlights the potential of developing these methods to predict the growth of coral reefs, both tropical and cold-water, under a variety of present and projected future conditions. To apply the model presented here in tropical systems, light as a variable would have to be included, with diurnal patterns and coral framework able to provide shade. Other key inclusions

would be generation of three-dimensional models, variable flow regimes (to better replicate *in situ* environments in different geographical regions), and additional growth and branching rules based on the success of tropical coral modelling studies [11,12,14].

Scaling up to reef-wide processes and using data from *in situ* systems would also increase the applicability of this model to *in situ* systems. The introduction of the death rule provides a way in which the importance of coral energetic reserves and physiological plasticity can be explored under various environmental conditions and includes the possibility to factor in events which may impact energetic reserves, such as reproduction or environmental stressors. Not only does the modelling of coral growth through the Goldilocks Principle open up new avenues for research and highlight the central rule that corals accrete by, but it also provides a further argument to the paradox of CWC distribution [32,35]. While *L. pertusa* reefs are often found in areas of high current velocity (10–50 cm s$^{-1}$ with peaks of up to 80–100 cm s$^{-1}$ [28–30,67,68]), they need flow velocities lower by an order of magnitude to successfully feed [32,35]. As demonstrated through SPH modelling of the Goldilocks Principle, growing coral framework modifies its local flow environment and reduces input velocities to speeds that are optimum for prey capture. In this way, corals are effectively optimizing their own local flow requirements through habitat engineering. By building on the way that corals can modify their own environment through SPH modelling coupled with the Goldilocks Principle, the methods outlined above give us a powerful tool to understand present-day and future coral reef growth, and provide an example of how simple rules can be used to understand and model ecological systems.

Data accessibility. All processed data are provided as part of the main text and electronic supplementary material. All raw data supporting the results are available through BODC [69].

Authors' contributions. S.J.H.: conceptualization, data curation, formal analysis, funding acquisition, investigation, methodology, project administration, writing—original draft, writing—review and editing; A.I.L.: formal analysis, investigation, methodology, resources, software, visualization, writing—original draft, writing—review and editing; C.O.: conceptualization, formal analysis, funding acquisition, investigation, methodology, resources, writing—original draft, writing—review and editing; A.G.: conceptualization, formal analysis, funding acquisition, investigation, methodology, writing—original draft, writing—review and editing; L.H.D.C.: formal analysis, investigation, methodology, writing—review and editing; Y.C.L.: conceptualization, investigation, methodology, resources, software, validation, visualization, writing—original draft, writing—review and editing; G.J.: formal analysis, investigation, methodology, software, validation, visualization; K.G.: formal analysis, investigation, methodology, resources, software, validation, visualization, writing—review and editing; N.A.K.: investigation, writing—review and editing; J.M.R.: conceptualization, funding acquisition, investigation, writing—review and editing.

All authors gave final approval for publication and agreed to be held accountable for the work performed therein.

Competing interests. We declare we have no competing interests.

Funding. This work was supported by the European Commission through the ASSEMBLE project EcoLophelia (grant agreement no. 227799) conducted in 2014 at the Sven Loven Centre for Marine Sciences-Tjärnö from the University of Gothenburg (Sweden). This work was supported by an NERC Doctoral Training Partnership (grant no. NE/L002558/1) to K.G., Independent Research Fellowships for N.A.K. and S.J.H. (NE/H010025, NE/K009028/1, NE/K009028/2) and the Royal Society of Edinburgh/Scottish Government to N.A.K. (RSE 48701/1). Funding to A.I.L. was supported by the Swedish Research Council FORMAS (grant no. 215-2012-1134). This paper is a contribution to the European Union's Horizon 2020 research and innovation programme under grant agreement no. 678760 (ATLAS) and no. 818123 (iAtlantic), and the UKRI GCRF One Ocean Hub (NE/S008950/1). It reflects the authors' views, and the European Union is not responsible for any use that may be made of the information it contains.

Acknowledgements. We thank the crew on board the RV Lophelia, especially Tomas Lundälv, as well as all personnel at Tjärnö who helped us during our stage at the Marine Laboratory.

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
