## [Peer Review File · Proceedings of the Royal Society B: Biological Sciences]

Review History

RSPB-2020-2562.R0 (Original submission)

Review form: Reviewer 1

Recommendation

Major revision is needed (please make suggestions in comments)

Scientific importance: Is the manuscript an original and important contribution to its field?

Acceptable

General interest: Is the paper of sufficient general interest?

Good

Quality of the paper: Is the overall quality of the paper suitable?

Good

Is the length of the paper justified?

No

Should the paper be seen by a specialist statistical reviewer?

No

Do you have any concerns about statistical analyses in this paper? If so, please specify them explicitly in your report.

No

It is a condition of publication that authors make their supporting data, code and materials available - either as supplementary material or hosted in an external repository. Please rate, if applicable, the supporting data on the following criteria.

Is it accessible?

N/A

Is it clear?

N/A

Is it adequate?

N/A

Do you have any ethical concerns with this paper?

No

Comments to the Author

The paper is about the role of complex three-dimensional CWC structures in the flow. To study this, data from Particle Imaging Velocimetry, and Smoothed Particle Hydrodynamic (SPH) was analyzed. The focus of the study was set on the role of "growth" affecting the hydrodynamics with regard to a model reef. The manuscript reads well, and is organized logically. Overall, I think the authors provided some very interesting results, in particular the PIV experiments look promising. In terms of the numerical model, I think there is not sufficient information (boundary conditions, time step, runtime, turbulence model, parameters e.g. viscosity, density ...) provided, so one can understand how the model works, how it was set up, and which experiments were conducted. I am sure that the findings described in the manuscript are of interest for the readership of proceedings B, but at this stage it is hard to judge due to the missing background information was actually done.

General remarks

There is no description about growth mechanism that causes the growth of the CWC. Looking at Figure 5 I can only guess that there must have been some kind of dynamic coupling implemented into the SPH environment that causes the growth of the CWCs. If so, please provide additional information about that. Moreover, I wonder to which amount does the insertion of new boundary particles (the yellow CWD particles in Figure 5) affect the pressure field, and hence the flow. What about numerical instabilities or shock waves, which have to be considered, while adding new particles into the system?

In the caption of Figure 5 there is a hint about the growing mechanism used. Here the authors state that the CWD was allowed to grow by one particle, per time step. I assume that the total runtime was 800s. I wonder to which amount does relate to any natural conditions, since the actual growth would be way too fast. However, in the discussion the author state that there is a need for of dynamic coupling, which is some sort of confusing.

The PIV plot indicates the flow field as one could expect from that experiment. However, comparing the field of the PIV experiment to the results of the numerical model one can get suspicious. Why is there no equivalent representation of the flow field that was computed by the SPH model? But in Figure 5 the flow field is color coded blue. A legend informing about the colors would also be helpful. I think that showing the changes of the flow field with regard to the growth of the CWC would allow for some sort of validation of the model. Moreover, the changes in the flow field caused by growing CWCs would be an interesting aspect for many researchers

working with CWCs.

There is no description about the turbulence model used. RANS / LES? I was wondering if the authors used some sort of pre-computed profile that was mapped on the inlet as described by Lund et al. (1998)? i.e. turbulence recycling method?

Was the model computed to steady state? Or at which time steps did the authors decide to analyze the flow field. How much time does the model actually represent? There is some sort of information indicated in Figure 5, but this is too little information to understand the experimental setup. A table would be helpful.

I could not find any reference that demonstrates the validity of the model. So, a comparison between the velocity profiles of the model that are plotted against those from the PIV experiment would be great to demonstrate the validity of the numerical model.

The use of dynamic boundary conditions is critical with regard to the number of used particles. A sensitivity study would be helpful to demonstrate that the number of particles is sufficient to resolve the hydrodynamics within the branches.

Was the SPH environment a software package or some sort of self-developed in house code. Unfortunately, I could not find any reference for that.

line by line comments:

line 154 Smoothed

line 160 “newly imposed boundary conditions”, what are they? Dynamic BC ? Reference needed

line 168 “unit of time” do you mean the time step, What is the time step ? Did the model converge ? What is the total runtime?

line 169 “The input velocity in all models was 50 cm/s to simulate a typically fast-moving current that CWC are subjected to”. This is not clear to me. At which location/position of your domain is the inlet? A more precise description about the model dimensions, boundary conditions, walls patches cyclic BCs is needed. There is an attempt in line 238 but this information is not sufficient to replicate the model.

line 216 Wendland - reference needed

line 247 Dynamic boundary conditions. Actually, these have been shown that an extremely high number of particles are needed to resolve the flow in an acceptable manner. Usually a gap develops between the fluid particles and the boundaries that is somewhat nonphysical. Anyway, 20,000 particles do not seem enough to actually resolve the flow around the branches of the CWC. Did the authors do some kind benchmark tests? Furthermore, I also could not find any attempt for a model validation.

line 284 “This highlights the need of a dynamic model such as SPH to simulate coral growth, as the environment”. This statement basically highlights that a way better description of the model in the methods section is needed, because it is not clear what was actually simulated and how the CWC grow.

line 346 “As demonstrated through SPH modelling of the Goldilocks Principle, growing coral framework modifies its local flow environment and reduces input velocities to speeds that are optimum for prey capture”. It is hard to judge about that statement, since no flow field was shown, and there is so little information about the experiments. I am missing a better description that describes what is actually happening in the SPH environment with regard to the growing

mechanism, and how the experiments were set up.

Figure 3, Over which timeframe were the flow fields averaged. Streamlines and flow vectors are shown, so please add streamlines.

Review form: Reviewer 2 (Di Tracey)

Recommendation

Accept with minor revision (please list in comments)

Scientific importance: Is the manuscript an original and important contribution to its field?

Good

General interest: Is the paper of sufficient general interest?

Excellent

Quality of the paper: Is the overall quality of the paper suitable?

Good

Is the length of the paper justified?

Yes

Should the paper be seen by a specialist statistical reviewer?

Yes

Do you have any concerns about statistical analyses in this paper? If so, please specify them explicitly in your report.

No

It is a condition of publication that authors make their supporting data, code and materials available - either as supplementary material or hosted in an external repository. Please rate, if applicable, the supporting data on the following criteria.

Is it accessible?

N/A

Is it clear?

N/A

Is it adequate?

N/A

Do you have any ethical concerns with this paper?

No

Comments to the Author

Excellent paper.

I have a few comments / requests of you that I have inserted into the attached pdf. I also made some editorial changes on the MS for you to consider

None of these comments are confidential. As I said to the publication editors, you are world leaders in deep-sea coral research and the MS will be of great value to the community.

Primarily I have asked that you:

in the 1st instance where you talk about the study species describe it fully, - 3D branching scleractinian stony coral

Define nubbin

Rewrite the distribution of this species as no, *Lophelia pertusa* is not found worldwide

Check when you mention of tropical corals that it is relevant and useful to do so and that it does not take away from the fact that this is a deep-sea coral study

Lines 284, 285 - the words in sentence 3 in the Discussion section do not help link the authors 1st 2x sentences - rewrite to make clear

The figure captions need expanding

there is an overuse of brackets

Decision letter (RSPB-2020-2562.R0)

08-Dec-2020

Dear Dr Hennige:

Your manuscript has now been peer reviewed and the reviews have been assessed by an Associate Editor. The reviewers' comments (not including confidential comments to the Editor) and the comments from the Associate Editor are included at the end of this email for your reference. As you will see, the reviewers and the Editors have raised some concerns with your manuscript and we would like to invite you to revise your manuscript to address them.

Research ethics:

Use of animals and field studies:

It is a condition of publication that you make available the data and research materials supporting the results in the article. Please see our Data Sharing Policies (<https://royalsociety.org/journals/authors/author-guidelines/#data>). Datasets should be deposited in an appropriate publicly available repository and details of the associated accession number, link or DOI to the datasets must be included in the Data Accessibility section of the article (<https://royalsociety.org/journals/ethics-policies/data-sharing-mining/>). Reference(s) to datasets should also be included in the reference list of the article with DOIs (where available).

Please submit a copy of your revised paper within three weeks. If we do not hear from you within this time your manuscript will be rejected. If you are unable to meet this deadline please let us know as soon as possible, as we may be able to grant a short extension.

Best wishes,
Dr Daniel Costa
mailto: proceedingsb@royalsociety.org

Associate Editor

Board Member: 1

Comments to Author:

The authors aim to address a gap related to modelling coral growth at an ecosystem scale, and they evaluate a new theoretical and methodological approach to address this limitation. This manuscript is clearly and concisely written, and the novel results would be of broad interest to the readership of Proceedings B once the details of the numerical model are repeatable and detailed in a sufficient manner.

Both referees find the study interesting and of general importance. Reviewer 1 had very minor edits and suggestions to improve the manuscript. Reviewer 2 requests further background information as this is needed to enable the reader to understand the numerical model and the specific experiments that were conducted. These revisions will ultimately allow a repeatable application of this novel modelling approach and improve the manuscript.

Based on the recommendation of the reviewers, a revised manuscript is requested that addresses both reviewer comments, with special attention to the request for clarity on the numerical model.

Reviewer(s)' Comments to Author:

Referee: 1

Comments to the Author(s)

The paper is about the role of complex three-dimensional CWC structures in the flow. To study this, data from Particle Imaging Velocimetry, and Smoothed Particle Hydrodynamic (SPH) was analyzed. The focus of the study was set on the role of "growth" affecting the hydrodynamics with regard to a model reef. The manuscript reads well, and is organized logically. Overall, I think the authors provided some very interesting results, in particular the PIV experiments look promising. In terms of the numerical model, I think there is not sufficient information (boundary conditions, time step, runtime, turbulence model, parameters e.g. viscosity, density ...) provided, so one can understand how the model works, how it was set up, and which experiments were conducted. I am sure that the findings described in the manuscript are of interest for the readership of proceedings B, but at this stage it is hard to judge due to the missing background information that was actually done.

General remarks

There is no description about growth mechanism that causes the growth of the CWC. Looking at Figure 5 I can only guess that there must have been some kind of dynamic coupling implemented into the SPH environment that causes the growth of the CWCs. If so, please provide additional information about that. Moreover, I wonder to which amount does the insertion of new boundary particles (the yellow CWD particles in Figure 5) affect the pressure field, and hence the flow. What about numerical instabilities or shock waves, which have to be considered, while adding new particles into the system?

In the caption of Figure 5 there is a hint about the growing mechanism used. Here the authors state that the CWD was allowed to grow by one particle, per time step. I assume that the total runtime was 800s. I wonder to which amount does relate to any natural conditions, since the actual growth would be way too fast. However, in the discussion the author state that there is a need for of dynamic coupling, which is some sort of confusing.

The PIV plot indicates the flow field as one could expect from that experiment. However, comparing the field of the PIV experiment to the results of the numerical model one can get suspicious. Why is there no equivalent representation of the flow field that was computed by the SPH model? But in Figure 5 the flow field is color coded blue. A legend informing about the

colors would also be helpful. I think that showing the changes of the flow field with regard to the growth of the CWC would allow for some sort of validation of the model. Moreover, the changes in the flow field caused by growing CWCs would be an interesting aspect for many researchers working with CWCs.

There is no description about the turbulence model used. RANS / LES? I was wondering if the authors used some sort of pre-computed profile that was mapped on the inlet as described by Lund et al. (1998)? i.e. turbulence recycling method?

Was the model computed to steady state? Or at which time steps did the authors decide to analyze the flow field. How much time does the model actually represent? There is some sort of information indicated in Figure 5, but this is too little information to understand the experimental setup. A table would be helpful.

I could not find any reference that demonstrates the validity of the model. So, a comparison between the velocity profiles of the model that are plotted against those from the PIV experiment would be great to demonstrate the validity of the numerical model.

The use of dynamic boundary conditions is critical with regard to the number of used particles. A sensitivity study would be helpful to demonstrate that the number of particles is sufficient to resolve the hydrodynamics within the branches.

Was the SPH environment a software package or some sort of self-developed in house code. Unfortunately, I could not find any reference for that.

line by line comments:

line 154 Smoothed

line 160 “newly imposed boundary conditions”, what are they? Dynamic BC ? Reference needed

line 168 “unit of time” do you mean the time step, What is the time step ? Did the model converge ? What is the total runtime?

line 169 “The input velocity in all models was 50 cm/s to simulate a typically fast-moving current that CWC are subjected to”. This is not clear to me. At which location/position of your domain is the inlet? A more precise description about the model dimensions, boundary conditions, walls patches cyclic BCs is needed. There is an attempt in line 238 but this information is not sufficient to replicate the model.

line 216 Wendland - reference needed

line 247 Dynamic boundary conditions. Actually, these have been shown that an extremely high number of particles are needed to resolve the flow in an acceptable manner. Usually a gap develops between the fluid particles and the boundaries that is somewhat nonphysical. Anyway, 20,000 particles do not seem enough to actually resolve the flow around the branches of the CWC. Did the authors do some kind benchmark tests? Furthermore, I also could not find any attempt for a model validation.

line 284 “This highlights the need of a dynamic model such as SPH to simulate coral growth, as the environment”. This statement basically highlights that a way better description of the model in the methods section is needed, because it is not clear what was actually simulated and how the CWC grow.

line 346 “As demonstrated through SPH modelling of the Goldilocks Principle, growing coral framework modifies its local flow environment and reduces input velocities to speeds that are optimum for prey capture”. It is hard to judge about that statement, since no flow field was shown, and there is so little information about the experiments. I am missing a better description

that describes what is actually happening in the SPH environment with regard to the growing mechanism, and how the experiments were set up.

Figure 3, Over which timeframe were the flow fields averaged. Streamlines and flow vectors are shown, so please add streamlines.

Referee: 2

Comments to the Author(s)

Excellent paper.

I have a few comments / requests of you that I have inserted into the attached pdf. I also made some editorial changes on the MS for you to consider

None of these comments are confidential. As I said to the publication editors, you are world leaders in deep-sea coral research and the MS will be of great value to the community.

Primarily I have asked that you:

in the 1st instance where you talk about the study species describe it fully, - 3D branching scleractinian stony coral

Define nubbin

Rewrite the distribution of this species as no, *Lophelia pertusa* is not found worldwide

Check when you mention of tropical corals that it is relevant and useful to do so and that it does not take away from the fact that this is a deep-sea coral study

Lines 284, 285 - the words in sentence 3 in the Discussion section do not help link the authors 1st 2x sentences - rewrite to make clear

The figure captions need expanding

there is an overuse of brackets

Author's Response to Decision Letter for (RSPB-2020-2562.R0)

See Appendix A.

RSPB-2020-2562.R1 (Revision)

Review form: Reviewer 1

Recommendation

Major revision is needed (please make suggestions in comments)

Scientific importance: Is the manuscript an original and important contribution to its field?

Good

General interest: Is the paper of sufficient general interest?

Good

Quality of the paper: Is the overall quality of the paper suitable?

Acceptable

Is the length of the paper justified?

Yes

Should the paper be seen by a specialist statistical reviewer?

No

Do you have any concerns about statistical analyses in this paper? If so, please specify them explicitly in your report.

No

It is a condition of publication that authors make their supporting data, code and materials available - either as supplementary material or hosted in an external repository. Please rate, if applicable, the supporting data on the following criteria.

Is it accessible?

Yes

Is it clear?

No

Is it adequate?

Yes

Do you have any ethical concerns with this paper?

No

Comments to the Author

This is my second review of the manuscript "Using the Goldilocks Principle to model coral ecosystem

Engineering" by Hennige, et al. As already pointed before the authors show interesting results with regard to the PIV measurements and the numerical model. The authors improved the manuscript significantly after the initial revision, which I appreciate a lot. I don't want to make an obsession out of it, but I am still not fully convinced about the feasibility of the SPH model.

Here are some examples:

- For my understanding it is good practice to provide sufficient information so anybody at least in theory would be able to replicate a numerical model. I am usually not picky, but one can only guess that water is simulated. For example, no information is given about the density, the viscosity, or the Reynolds number. By which routine were the particles generated. Other basic information such as the kernel, the viscosity model, speed of sound are missing. I miss a table that summarizes all parameters that are essential to get the model running. There is no information about the model itself. For example, who developed it, or was it validated at some point.

- The patches within the flow field of the supplementary figure usually indicate numerical pressure fluctuations and indicate that the speed of sound may not be chosen adequately. One should provide this parameter. Where there any corrective methods applied like deltaSPH or a shifting algorithm?

- There is no sufficient information about the mathematical algorithms that were implemented to control, growth and death of the corals. In SPH each particle has a specific coordinate, as well as a set of parameters. If one inserts or removes particles from the system, this needs to be done in some sort of geometric arrangement, and needs to be controlled by some kind of mathematical algorithm, which is not described here. Moreover, it is not clear by which criterion the geometric arrangement (stacking of solid particles) is controlled? There is not a single sentence or equation given that allows to understand in which way, particles are added or removed from the system. At a minimum a reference is needed, to provide the necessary information that controls this complex process.

Review form: Reviewer 2 (Di Tracey)

Recommendation

Accept as is

Scientific importance: Is the manuscript an original and important contribution to its field?

Excellent

General interest: Is the paper of sufficient general interest?

Excellent

Quality of the paper: Is the overall quality of the paper suitable?

Good

Is the length of the paper justified?

Yes

Should the paper be seen by a specialist statistical reviewer?

No

Do you have any concerns about statistical analyses in this paper? If so, please specify them explicitly in your report.

No

It is a condition of publication that authors make their supporting data, code and materials available - either as supplementary material or hosted in an external repository. Please rate, if applicable, the supporting data on the following criteria.

Is it accessible?

Yes

Is it clear?

Yes

Is it adequate?

Yes

Do you have any ethical concerns with this paper?

No

Comments to the Author

Thanks for acting on my comments, the paper reads very well, congratulations.

Decision letter (RSPB-2020-2562.R1)

06-Mar-2021

Dear Dr Hennige:

I am writing to inform you that your manuscript # RSPB-2020-2562.R1 entitled "Using the Goldilocks Principle to model coral ecosystem engineering" has been rejected for publication in Proceedings B.

This action has been taken on the advice of referees, who have recommended that substantial revisions are necessary. We have chosen a rejection and resubmit rather than revision, as this will allow you more time to carry out the analysis requested by the referee. With this in mind we would be happy to consider a resubmission, provided the comments of the referees are fully addressed. However please note that this is not a provisional acceptance.

Please find below the comments made by the referees, not including confidential reports to the Editor, which I hope you will find useful.

- 1) A 'response to referees' document including details of how you have responded to the comments, and the adjustments you have made.
- 2) A clean copy of the manuscript and one with 'tracked changes' indicating your 'response to referees' comments document.
- 3) Line numbers in your main document.
- 4) Please read our Data sharing policies to ensure that you meet our requirements (<https://royalsociety.org/journals/authors/author-guidelines/#data>).

Sincerely,
Dr Daniel Costa
Editor, Proceedings B
proceedingsb@royalsociety.org

Reviewer(s)' Comments to Author:

Referee: 1

Comments to the Author(s)

This is my second review of the manuscript "Using the Goldilocks Principle to model coral ecosystem

Engineering" by Hennige, et al. As already pointed before the authors show interesting results with regard to the PIV measurements and the numerical model. The authors improved the manuscript significantly after the initial revision, which I appreciate a lot. I don't want to make an obsession out of it, but I am still not fully convinced about the feasibility of the SPH model.

Here are some examples:

- For my understanding it is good practice to provide sufficient information so anybody at least in theory would be able to replicate a numerical model. I am usually not picky, but one can only guess that water is simulated. For example, no information is given about the density, the viscosity, or the Reynolds number. By which routine were the particles generated. Other basic information such as the kernel, the viscosity model, speed of sound are missing. I miss a table that summarizes all parameters that are essential to get the model running. There is no information about the model itself. For example, who developed it, or was it validated at some point.

- The patches within the flow field of the supplementary figure usually indicate numerical pressure fluctuations and indicate that the speed of sound may not be chosen adequately. One should provide this parameter. Where there any corrective methods applied like deltaSPH or a shifting algorithm?

- There is no sufficient information about the mathematical algorithms that were implemented to control, growth and death of the corals. In SPH each particle has a specific coordinate, as well as a set of parameters. If one inserts or removes particles from the system, this needs to be done in some sort of geometric arrangement, and needs to be controlled by some kind of mathematical algorithm, which is not described here. Moreover, it is not clear by which criterion the geometric arrangement (stacking of solid particles) is controlled? There is not a single sentence or equation given that allows to understand in which way, particles are added or removed from the system. At a minimum a reference is needed, to provide the necessary information that controls this complex process.

Referee: 2

Comments to the Author(s)

Thanks for acting on my comments, the paper reads very well, congratulations.

Author's Response to Decision Letter for (RSPB-2020-2562.R1)

See Appendix B.

RSPB-2021-1260.R0

Review form: Reviewer 2 (Di Tracey)

Recommendation

Accept with minor revision (please list in comments)

Scientific importance: Is the manuscript an original and important contribution to its field?

Excellent

General interest: Is the paper of sufficient general interest?

Excellent

Quality of the paper: Is the overall quality of the paper suitable?

Good

Is the length of the paper justified?

Yes

Should the paper be seen by a specialist statistical reviewer?

Yes

Do you have any concerns about statistical analyses in this paper? If so, please specify them explicitly in your report.

No

It is a condition of publication that authors make their supporting data, code and materials available - either as supplementary material or hosted in an external repository. Please rate, if applicable, the supporting data on the following criteria.

Is it accessible?

N/A

Is it clear?

N/A

Is it adequate?

N/A

Do you have any ethical concerns with this paper?

No

Comments to the Author

This paper is of a very high standard, the results very useful for deepsea coral researchers, particularly the data presented describing how corals regulate local flow will be applicable and useful. My edits highlighted as comments in the attached, are for consideration, not all are vital.

At times a little more detail is required e.g., add in the word coral more regularly, to make sure the reader knows what is being referred to, use the words flume tank not flumes, describe what 'this' is, and so on. The Goldilocks Principle description could be replaced with the words 'just right environment' on occasion to avoid repetition. At times some ideas need to be linked or examples given.

Don't hesitate to reach out if any comments are unclear.

Decision letter (RSPB-2021-1260.R0)

05-Jul-2021

Dear Dr Hennige

I am pleased to inform you that your manuscript RSPB-2021-1260 entitled "Using the Goldilocks Principle to model coral ecosystem engineering" has been accepted for publication in Proceedings B.

The referee(s) have recommended publication, but also suggest some minor revisions to your manuscript. Therefore, I invite you to respond to the referee(s)' comments and revise your manuscript. Because the schedule for publication is very tight, it is a condition of publication that

you submit the revised version of your manuscript within 7 days. If you do not think you will be able to meet this date please let us know.

[http://datadryad.org/submit?journalID=RSPB&manu=\(Document not available\)](http://datadryad.org/submit?journalID=RSPB&manu=(Document+not+available)) which will take you to your unique entry in the Dryad repository. If you have already submitted your data to dryad you can make any necessary revisions to your dataset by following the above link. Please see <https://royalsociety.org/journals/ethics-policies/data-sharing-mining/> for more details.

Sincerely,

Dr Daniel Costa

Associate Editor

Board Member

Comments to Author:

This manuscript is of a very high standard and the results are relevant to the coral research global community. However, there is additional detail required to ensure the approach is clear and the methods are repeatable. Please carefully consider the reviewer's highlighted comments to further improve the manuscript.

Reviewer(s)' Comments to Author:

Referee: 2

Comments to the Author(s).

This paper is of a very high standard, the results very useful for deepsea coral researchers, particularly the data presented describing how corals regulate local flow will be applicable and useful. My edits highlighted as comments in the attached, are for consideration, not all are vital.

At times a little more detail is required e.g., add in the word coral more regularly, to make sure the reader knows what is being referred to, use the words flume tank not flumes, describe what 'this' is, and so on. The Goldilocks Principle description could be replaced with the words 'just right environment' on occasion to avoid repetition. At times some ideas need to be linked or examples given.

Don't hesitate to reach out if any comments are unclear.

Author's Response to Decision Letter for (RSPB-2021-1260.R0)

See Appendix C.

Decision letter (RSPB-2021-1260.R1)

15-Jul-2021

Dear Dr Hennige

I am pleased to inform you that your manuscript entitled "Using the Goldilocks Principle to model coral ecosystem engineering" has been accepted for publication in Proceedings B.

Data Accessibility section

Open Access

Paper charges

Sincerely,

Proceedings B

Appendix A

Dr Daniel Costa
mailto:proceedingsb@royalsociety.org

Associate Editor

Board Member: 1

Comments to Author:

The authors aim to address a gap related to modelling coral growth at an ecosystem scale, and they evaluate a new theoretical and methodological approach to address this limitation. This manuscript is clearly and concisely written, and the novel results would be of broad interest to the readership of Proceedings B once the details of the numerical model are repeatable and detailed in a sufficient manner.

Both referees find the study interesting and of general importance. Reviewer 1 had very minor edits and suggestions to improve the manuscript. Reviewer 2 requests further background information as this is needed to enable the reader to understand the numerical model and the specific experiments that were conducted. These revisions will ultimately allow a repeatable application of this novel modelling approach and improve the manuscript.

Based on the recommendation of the reviewers, a revised manuscript is requested that addresses both reviewer comments, with special attention to the request for clarity on the numerical model.

The authors thank the reviewers and editor for assessing this manuscript. We have detailed our responses below to clarify our methods, and to address other minor concerns.

Referee: 1

Comments to the Author(s)

The paper is about the role of complex three-dimensional CWC structures in the flow. To study this, data from Particle Imaging Velocimetry, and Smoothed Particle Hydrodynamic (SPH) was analyzed. The focus of the study was set on the role of “growth” affecting the hydrodynamics with regard to a model reef. The manuscript reads well, and is organized logically. Overall, I think the authors provided some very interesting results, in particular the PIV experiments look promising. In terms of the numerical model, I think there is not sufficient information (boundary conditions, time step, runtime, turbulence model, parameters e.g. viscosity, density ...) provided, so one can understand how the model works, how it was set up, and which experiments were conducted. I am sure that the findings described in the manuscript are of interest for the readership of proceedings B, but at this stage it is hard to judge due to the missing background information that was actually done.

We thank the referee for their comments and have now included a lot more detail with regard to the points, detailed below.

General remarks

There is no description about growth mechanism that causes the growth of the CWC. Looking at Figure 5 I can only guess that there must have been some kind of dynamic coupling implemented into the SPH environment that causes the growth of the CWCs. If so, please provide additional information about that.

Additional details of the growth mechanism employed to model the CWC have now been added (see paragraph beginning with “The seabed and coral...”) to better describe the modelling process.

Moreover, I wonder to which amount does the insertion of new boundary particles (the yellow CWD particles in Figure 5) affect the pressure field, and hence the flow. What about numerical instabilities or shock waves, which have to be considered, while adding new particles into the system?

This is a good question - A description of how the coral particles are added to mimic coral growth is now included in line (303) (see also comment above). New particles were not inserted into the domain but fluid particles were swapped out for coral ones to help preserve stability and avoid potential problems as indicated in the comment above.

In the caption of Figure 5 there is a hint about the growing mechanism used. Here the authors state that the CWD was allowed to grow by one particle, per time step. I assume that the total runtime was 800s.

*A paragraph that better describes the coral growth mechanism is now provided in the manuscript. It is correct (as indicated in Fig 5) that for the set of simulated results presented, the coral is allowed to grow at each time-step provided that the surrounding hydrodynamics meets the growth rule. The coral’s **maximum** growth potential is capped at $1.5 \times Dx$ in each direction (so essentially one particle in each direction) but the actual growth would depend on the flow conditions.*

I wonder to which amount does relate to any natural conditions, since the actual growth would be way too fast. However, in the discussion the author state that there is a need for of dynamic coupling, which is some sort of confusing.

The results presented here aim to show the potential of the SPH methodology and possibilities of modelling long time-scale coral growth predictions using projected conditions. The model demonstrated the growth of a theoretical coral framework based on simple principles based on the Goldilocks rules. The statement on dynamic coupling in the discussion has been removed to avoid confusion, and we have clarified in the introduction and methods that this growth is a theoretical simulation not using real growth rates.

The PIV plot indicates the flow field as one could expect from that experiment. However, comparing the field of the PIV experiment to the results of the numerical model one can get suspicious. Why is there no equivalent representation of the flow field that was computed by the SPH model?

The aim of the PIV results (now paragraph starting at line 143) was to quantify the flow around corals in a variety of scenarios in the view to understand how coral framework can modify flow environment within a reef, while the SPH simulated results predict how coral will grow when given simple rules that follow the Goldilocks Principle. Qualitative comparison showed that the developed SPH model not only confirmed coral growth through the Goldilocks Principle, but opens up new avenues for research and highlights the central rule that corals accrete by, providing further evidence to the paradox of CWC distribution with regard to flow speed.

But in Figure 5 the flow field is color coded blue. A legend informing about the colors would also be helpful. I think that showing the changes of the flow field with regard to the growth of the CWC would allow for some sort of validation of the model. Moreover, the changes in the flow field caused by growing CWCs would be an interesting aspect for many researchers working with CWCs.

We agree with the reviewer and have added several pieces of information around this point. We have clarified in the results that these are not PIV results, and that the blue relates to water particles in the model. However, to build on this, we have included a new supplemental figure (and text in results) of the SPH model with streamlines (Supplementary Figure 3) to demonstrate the flow fields as would be visualised using PIV. We have further added to this by highlighting areas which would promote coral growth as according to the Goldilocks Rule applied in this manuscript.

There is no description about the turbulence model used. RANS / LES? I was wondering if the authors used some sort of pre-computed profile that was mapped on the inlet as described by Lund et al. (1998)? i.e. turbulence recycling method?

Additional details have been added to the manuscript to clarify that no turbulence model was employed, and that the SPH solver solves both the mass and momentum conservation equations with dynamic boundary conditions applied on solid particles, which encompass the seabed and coral. Additional physical growth attributes are specified for the coral particles.

Was the model computed to steady state? Or at which time steps did the authors decide to analyze the flow field. How much time does the model actually represent? There is some sort of information indicated in Figure 5, but this is too little information to understand the experimental setup. A table would be helpful.

This has now been clarified in the text and figure captions - one of the conditions for initiating coral growth is that the local velocities around the corals are steady. It should be noted that the time scale of the model does not represent real physical time but a means to advance the simulation iteratively on successive time-steps. This is now clarified in line (376).

I could not find any reference that demonstrates the validity of the model. So, a comparison between the velocity profiles of the model that are plotted against those from the PIV experiment would be great to demonstrate the validity of the numerical model.

We have included a new reference to clarify this, with more information also included for interested readers to explore the validity and accuracy of the numerical solver. The new references demonstrate the validity of the methodology and how SPH methodology has been used to model many fluid flow problems in engineering. Additionally, the new reference with the same SPH model and solver was used and validated against cross-discipline engineering problems in pharmaceutical oscillatory baffled reactors where detailed investigation of velocity profiles was measured and benchmarked against commercial CFD solvers (ANSYS Fluent). [Jimeno Millor, G., Lee, Y. C., & Ni, X-W. \(2019\). Smoothed Particle Hydrodynamics - A New Approach for Modeling Flow in Oscillatory Baffled Reactors. Computers and Chemical Engineering, 124, 14-27. https://doi.org/10.1016/j.compchemeng.2019.02.003](https://doi.org/10.1016/j.compchemeng.2019.02.003)

The use of dynamic boundary conditions is critical with regard to the number of used particles. A sensitivity study would be helpful to demonstrate that the number of particles is sufficient to resolve the hydrodynamics within the branches.

It is now clarified in the text that the dynamic boundary conditions for the present study are not significantly affected by the number of particles employed in the system. This is due to the nature of the slow-moving flow typically observed in the vicinity of the coral branches, and the fact that the SPH kernel at these locations have a full set of neighbouring particles which negates one of the main drawbacks of using dynamic boundaries.

Was the SPH environment a software package or some sort of self-developed in house code. Unfortunately, I could not find any reference for that.

Additional details of the SPH solver are now provided in the text with note to a new reference regarding the numerical details used in the present solver. The SPH solver was developed in-house and benchmarked in Jimeno Millor et al. 2019.

line by line comments:

line 154 Smoothed

Now changed. Thank you.

line 160 “newly imposed boundary conditions”, what are they? Dynamic BC ? Reference needed

This has now been clarified in the text - they refer to particles changing from fluid to solid ones during coral growth.

line 168 “unit of time” do you mean the time step, What is the time step? Did the model converge ? What is the total runtime?

“Unit of time” and “Time-step” have now been replaced by ‘growth-step’ to clarify that we are not replicating real world time scenarios.

line 169 “The input velocity in all models was 50 cm/s to simulate a typically fast-moving current that CWC are subjected to”. This is not clear to me. At which location/position of your domain is the inlet? A more precise description about the model dimensions, boundary conditions, walls patches cyclic BCs is needed. There is an attempt in line 238 but this information is not sufficient to replicate the model.

We have now clarified (and expanded) upon this description in the SPH methodology section to ensure this is replicable.

line 216 Wendland - reference needed

2 new references have now been added: F. Macia, M. Antuono, and A. Colagrossi. Benefits of using a Wendland kernel for free-surface flows. 6th International SPHERIC workshop, Hamburg, Germany, 2011, and H. Wendland. Piecewise polynomial, positive definite and compactly supported radial functions of minimal degree. Adv. Comput. Math., 4:389–396, 1995

line 247 Dynamic boundary conditions. Actually, these have been shown that an extremely

high number of particles are needed to resolve the flow in an acceptable manner. Usually a gap develops between the fluid particles and the boundaries that is somewhat nonphysical. Anyway, 20,000 particles do not seem enough to actually resolve the flow around the branches of the CWC. Did the authors do some kind benchmark tests? Furthermore, I also could not find any attempt for a model validation.

The reviewer is correct that for dynamic boundary conditions, the aforementioned problem typically arise. However, the flow conditions for coral growth is relatively small and the nature of distribution of coral particles help negates these effects. In addition, the problem solved is 2-dimensional, and the goal of the numerical study here is to provide a macroscopic view of the coral. With regard to the benchmarks, please see above comments.

line 284 “This highlights the need of a dynamic model such as SPH to simulate coral growth, as the environment”. This statement basically highlights that a way better description of the model in the methods section is needed, because it is not clear what was actually simulated and how the CWC grow.

Agreed, and this has been expanded upon in the Methods section.

line 346 “As demonstrated through SPH modelling of the Goldilocks Principle, growing coral framework modifies its local flow environment and reduces input velocities to speeds that are optimum for prey capture”. It is hard to judge about that statement, since no flow field was shown, and there is so little information about the experiments. I am missing a better description that describes what is actually happening in the SPH environment with regard to the growing mechanism, and how the experiments were set up.

In addition to expanding and clarifying methodology used, we have included additional supplementary figures (Supplementary Figure 3), which demonstrates the reduction of flow within SPH models.

Figure 3, Over which timeframe were the flow fields averaged. Streamlines and flow vectors are shown, so please add streamlines.

The figure has been updated to indicate ‘flow vectors and streamlines’ and detail has been added to the legend that average flow velocity vector fields and velocity magnitudes were from 2-minute PIV recordings at 10 Hz.

Referee: 2

Comments to the Author(s)

Excellent paper.

I have a few comments / requests of you that I have inserted into the attached pdf. I also made some editorial changes on the MS for you to consider

Many thanks for the reviewer for their positive comments – we have addressed the points raised in the PDF and below with expanded or clarified text and accepting suggestions.

None of these comments are confidential. As I said to the publication editors, you are world leaders in deep-sea coral research and the MS will be of great value to the community.

Primarily I have asked that you:

- in the 1st instance where you talk about the study species describe it fully, - 3D branching scleractinian stony coral
 - This has been clarified as suggested
- Define nubbin
 - This has been clarified as small fragments with ~ 5 polyps.
- Rewrite the distribution of this species as no, *Lophelia pertusa* is not found worldwide
 - The referee is correct and this has now been clarified
- Check when you mention of tropical corals that it is relevant and useful to do so and that it does not take away from the fact that this is a deep-sea coral study
 - Agreed – this has been checked and amended where necessary. While parallels are drawn to the use that such an SPH model could have for tropical corals, the focus has been checked.
- Lines 284, 285 – the words in sentence 3 in the Discussion section do not help link the authors 1st 2x sentences - rewrite to make clear
 - This has been reworded to clarify
- The figure captions need expanding
 - These have been expanded to clarify the contents of figures
- there is an overuse of brackets
 - Agreed, and these have been reduced

Appendix B

This manuscript is a resubmission. Original submission ID is RSPB-2020-2562.R1

Response to Reviewers

Dear Dr. Costa,

Many thanks to yourself and the Reviewers for re-reviewing our Manuscript “*Using the Goldilocks Principle to model coral ecosystem engineering*”. Reviewer 1 highlighted some additional points regarding the types of model which we used, and we welcome the opportunity to strengthen our manuscript by addressing their points. We have now included additional text, equations, references and a ‘pseudocode’ which is now included in the supplementary materials to help rectify this. More specific responses are addressed inline below.

Reviewer 1

For my understanding it is good practice to provide sufficient information so anybody at least in theory would be able to replicate a numerical model. I am usually not picky, but one can only guess that water is simulated. For example, no information is given about the density, the viscosity, or the Reynolds number. By which routine were the particles generated. Other basic information such as the kernel, the viscosity model, speed of sound are missing. I miss a table that summarizes all parameters that are essential to get the model running. There is no information about the model itself. For example, who developed it, or was it validated at some point.

We agree with the reviewer and have clarified these points. We have added a supplementary table which includes information on the type of fluid (water), density, viscosity and reference speed of sound employed in the simulation. In the table we have included information about initial particle separation and smoothing length to support the reviewer’s comments regarding the kernel used. Within the text, we have clarified our use of viscosity (that we are not using an artificial viscosity model), and have included references to those we use (Morris 1997). To clarify our approach, we have re-arranged the equations of motion within the text to better support the narrative explanation. The fluid particles in the domain were generated manually within the developed SPH solver; the SPH solver was developed in-house which was extended from another inter-disciplinary research work (ref 46) and is cited in the text. This solver has been extensively tested and validated against ANSYS Fluent CFD on mixing in baffled reactors on transport of multi-species flows.

The patches within the flow field of the supplementary figure usually indicate numerical pressure fluctuations and indicate that the speed of sound may not be chosen adequately. One should provide this parameter. Where there any corrective methods applied like deltaSPH or a shifting algorithm?

We have now included additional text (with parameters and equations) within the main text to clarify this point, and to highlight our density smoothing approach. We kept the SPH algorithm as simple and pure as possible without relying on corrective methods such as delta-SPH or particle shifting methods.

There is no sufficient information about the mathematical algorithms that were implemented to control, growth and death of the corals. In SPH each particle has a specific coordinate, as well as a set of parameters. If one inserts or removes particles from the system, this needs to

be done in some sort of geometric arrangement, and needs to be controlled by some kind of mathematical algorithm, which is not described here. Moreover, it is not clear by which criterion the geometric arrangement (stacking of solid particles) is controlled? There is not a single sentence or equation given that allows to understand in which way, particles are added or removed from the system. At a minimum a reference is needed, to provide the necessary information that controls this complex process.

The authors agree that this is a complex process and we have taken several steps to clarify our approach in the amended manuscript. We have changed our wording to clarify that we are not creating new coral particles and that no new particles are added to the domain, but instead fluid particles are converted into coral particles. To support this further, we have now included a 'pseudocode' in the supplementary material, that illustrates the model workflow of converting fluid particles to a coral particle, and the conditions under which they could then convert to dead coral particles.

Appendix C

Response to referees

Associate Editor

Board Member

Comments to Author:

This manuscript is of a very high standard and the results are relevant to the coral research global community. However, there is additional detail required to ensure the approach is clear and the methods are repeatable. Please carefully consider the reviewer's highlighted comments to further improve the manuscript.

We thank the editor for these comments and have address all remaining concerns of the referee below. We feel that the revised manuscript is much improved on the original submission.

Referee: 2

Comments to the Author(s).

This paper is of a very high standard, the results very useful for deepsea coral researchers, particularly the data presented describing how corals regulate local flow will be applicable and useful. My edits highlighted as comments in the attached, are for consideration, not all are vital.

At times a little more detail is required e.g., add in the word coral more regularly, to make sure the reader knows what is being referred to, use the words flume tank not flumes, describe what 'this' is, and so on. The Goldilocks Principle description could be replaced with the words 'just right environment' on occasion to avoid repetition. At times some ideas need to be linked or examples given.

We thank the referee for re-reviewing this manuscript and for their insightful comments at each stage. We have actioned the remaining suggestions from the annotated PDF, with changes of phrasing, additional information, and some clarifications.